# Rapid determination of seismic influence field based on mobile communication big data—A case study of the Luding Ms 6.8 earthquake in Sichuan, China

**Dongping Li[1], Qingquan Tan[2]\*, Zhiyi Tong[3], Jingfei Yin[1], Min Li[1], Huanyu Li[1], Haiqing Sun[1]**

**1** Zhejiang Earthquake Agency, Hangzhou, China, **2** Beijing Earthquake Agency, Beijing, China, **3** Zhejiang Development and Planning Institute, Hangzhou, China

\* earthquake-yingji@zjdz.gov.cn

**Data Availability Statement:** All relevant data are within the paper and its Supporting Information files.

## Abstract

Smartphone location data provide the most direct field disaster distribution data with low cost and high coverage. The large-scale continuous sampling of mobile device location data provides a new way to estimate the distribution of disasters with high temporal–spatial resolution. On September 5, 2022, a magnitude 6.8 earthquake struck Luding County, Sichuan Province, China. We quantitatively analyzed the Ms 6.8 earthquake from both temporal and geographic dimensions by combining 1,806,100 smartphone location records and 4,856 spatial grid locations collected through communication big data with the smartphone data under 24-hour continuous positioning. In this study, the deviation of multidimensional mobile terminal location data is estimated, and a methodology to estimate the distribution of out-of-service communication base stations in the disaster area by excluding micro error data users is explored. Finally, the mathematical relationship between the seismic intensity and the corresponding out-of-service rate of communication base stations is established, which provides a new technical concept and means for the rapid assessment of post-earthquake disaster distribution.

## Introduction

For a long time, most of the disaster distribution data related to post-earthquake rescue has been obtained by using expensive special equipment. However, due to the uncertainty of when and where an earthquake disaster may occur, the large-scale deployment and maintenance of professional equipment can incur considerable costs. It is also difficult to ensure the connectivity of this equipment and the full coverage of all affected people when an earthquake occurs [1]. With their ever-increasing popularity, smartphones, as the most widely used electronic devices, have been equipped with computing, communication, storage, and sensing capabilities. Even in disaster scenarios, the probability of people holding smartphones is still very high. Therefore, smartphones are capable of constructing direct field disaster distribution data with

**Funding:** This study was supported by the Scientific Research Fund of Institute of Engineering Mechanics, China Earthquake Administration (2021D07), Project of Spark Program of Earthquake Sciences, China Earthquake Administration (XH23001B), Zhejiang Provincial Natural Science Foundation of China (LTGG24D040002).The funders had no role in study design, data collection and analysis, decision to publish, or preparation of the manuscript.

**Competing interests:** The authors have declared that no competing interests exist.

low cost and high coverage. The large-scale continuous sampling of mobile device location data provides a new way to estimate disaster distribution with higher temporal–spatial resolution [2].

Several earthquakes in recent years have shown that when the intensity of the epicenter reaches VIII, communication base stations will usually be out of service, which directly leads to a precipitous drop in the acquisition of smartphone location data after an earthquake. A large amount of smartphone location data disappears at a large scale after an earthquake, and the closer to the hardest hit area, the more obvious the data drop is. In addition, in the event of a nondestructive earthquake, there will be such phenomena as an increase in communication volume at the epicenter and location changes due to the flow of people avoiding the disaster. The change and distribution of mobile communication big data play an indicative role in estimating the extent of devastation in the first instances [3]. After the Wenchuan earthquake in 2008, communication facilities in the disaster area were severely damaged, and many mobile base stations stopped service, resulting in communication outages in the areas where these stations are located. We analyzed the affected areas and the extent of devastation by collecting the out-of-service data of mobile base stations and mapped the distribution range of the affected areas, which was highly consistent with the intensity distribution data obtained from the post-earthquake field survey (Fig 1).

(We have collected information after the Wenchuan Ms 8.0 earthquake through 2 specialized BBS forums: https://www.txrjy.com/; https://club.mscbsc.com/. Although the data we collected are incomplete and limited, we still have relatively and accurately demonstrated the distribution of the seismic influence field and the approximate area of the macroscopic epicenter through spatial interpolation).

With the prevalence of smartphones and the development of mobile Internet services, in combination with the popularization of global positioning technology, the technology of population data estimation based on precise geographic location has become increasingly mature [4]. When a user enters a certain geographic space, it is possible to obtain and verify the user's location information, and the current population size of a geographic space can be inferred through a differentiated model. After an earthquake, we can use the changes in smartphone thermal data to infer the extent of damage to mobile communication infrastructure caused by the earthquake [5,6]. The data sources involved in this work are the various smartphone APP vendors. Because the data is location data of point groups reported in a certain area, personal privacy is not involved, and the data size covers billions of terminals.

Since 2012, telecom operators have successively applied location data analysis to mobile networks. Verizon, a U.S. operator, engages in business consulting by collecting information on the apps used and websites visited by its users, as well as their geographic locations [7]. The highway monitoring project of French telecom operator Orange and the smart footprint project of Spanish telecom company Telefonica are designed to provide location-derived information to users. Represented by the I-LOV project in Germany, many research institutions have participated in the construction of disaster emergency rescue systems based on smartphone signal searches [8]. At the World Internet Conference in Wuzhen in November 2014, the comprehensive analysis of China Mobile big data demonstrated a dynamic people flow big data analysis platform, and in December of the same year, the mobile location of big data provided decision-making information for the government in the emergency response to the stampede at the Bund in Shanghai [9,10]. "Location big data" is not merely the result of technological transformation in the computer industry. Cross-border thinking and big data thinking are also applicable to various other industries. Similar work has been conducted in the field of natural disaster research, where MyShake has been developed and built as a global smartphone seismic network that can detect and be triggered by P waves. With the constant downloading

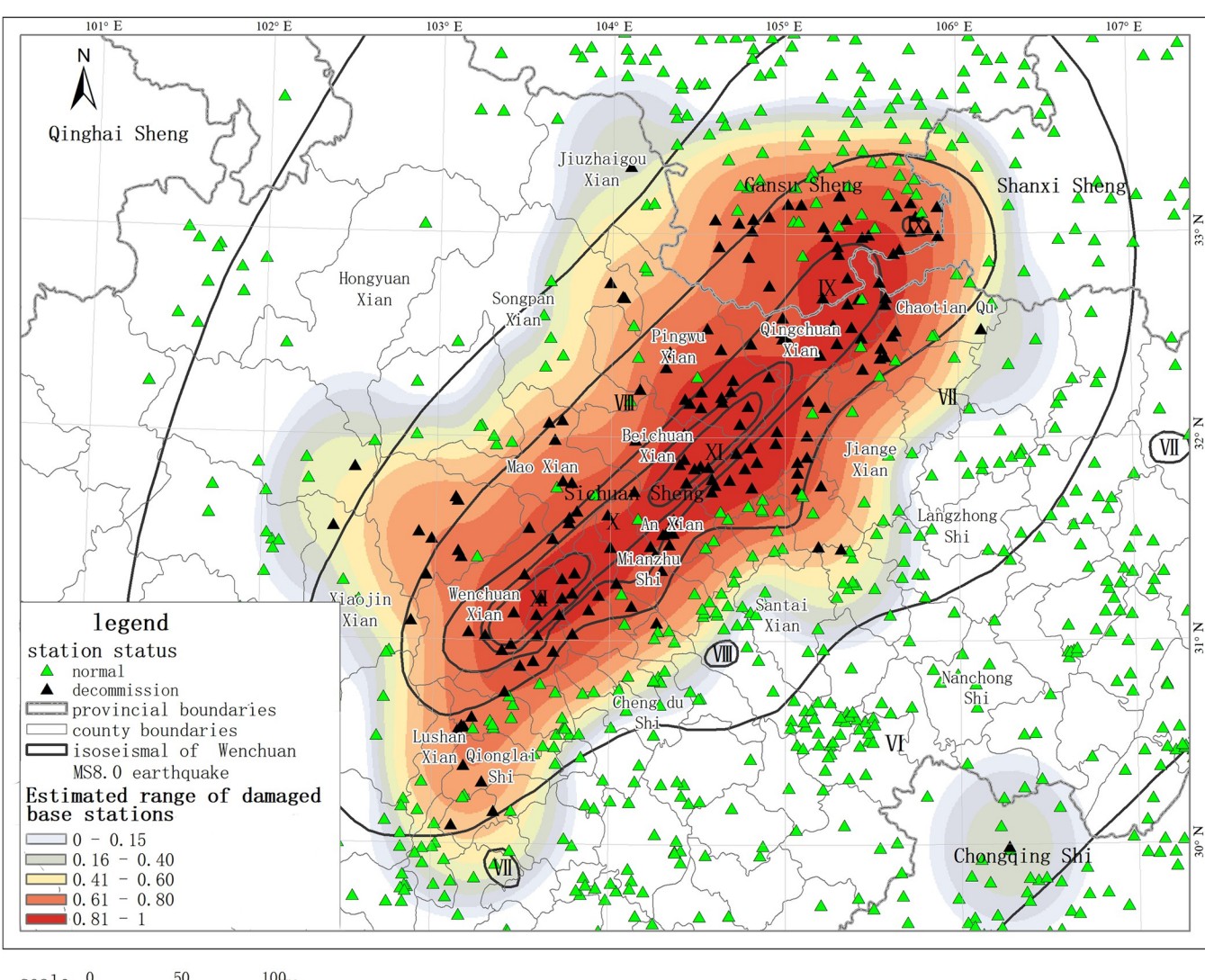

**Fig 1. Distribution of out-of-service base stations and intensity after Wenchuan Ms 8.0 Earthquake.**

of MyShake, the scale of the seismic network has been rapidly developed, forming a global seismic network that utilizes personal smartphones to provide acceleration waveforms [11]. Since 2017, experts have started participating in the research on rapid acquisition of disaster situations and post-earthquake crowd flow analysis based on mobile Internet location data, and some scholars have begun using mobile location information data to study and track the distribution of earthquake-stricken areas and the responses of people after the earthquake [12]. Researchers have used smartphone data to analyze the crowd dynamics during the 2017 Jiuzhaigou earthquake. By analyzing the call and SMS data, they identified the densely populated areas and the migration trajectories during the evacuation process, which provided support for the development of effective urgent evacuation strategies [13]. The MIT Media Lab used smartphone data to analyze changes in human mobility and access to critical urban services in the aftermath of the 2015 Nepal earthquake, and the findings underscored the importance of considering post-disaster mobility dynamics in emergency response and recovery planning

[14]. European-Mediterranean Seismological Centre used smartphone data to dynamically assess post-earthquake population displacement. By analyzing detailed call records [15], EMSC identified changes in human mobility patterns in the affected areas and quantified the displacement level, which provided information for post-disaster recovery and planning work. Taking an earthquake-stricken area as the main research object, Gao Na compared the demographic data obtained by smartphone location and found the role of annual difference data in the field of sudden disaster emergency relief [16]. The Nie group from the Institute of Geology, China Earthquake Administration used smartphone location data to analyze the indicators related to earthquake disasters and proposed to regard seismic intensity as a sensitive indicator for smartphone location data [17]. Zhang used the pre-earthquake and post-earthquake Internet smartphone location data in an earthquake area and adopted the standard deviation ellipse model in spatial econometric analysis to analyze the spatial distribution characteristics of the disappeared smartphone location data in the earthquake area and the oriented direction of its discrete point sets, which determined the direction of the seismic influence field, and further provided technical support for post-disaster situation assessment and emergency response services [18]. In the process of seismic data processing, relevant experts can address earthquake emergency response, rescue guidance, and other studies by integrating and mining a large amount of complex and multisource data [19]. In conclusion, the application of "mobile location big data" during earthquake emergency responses can improve the scientificity and accuracy of the decision-making processes during an earthquake and enhance the capabilities of earthquake early warnings and emergency responses [20].

## Materials and methods

### Principles of smartphone location big data

The popularity of smartphones and the development of the Internet (according to the report of "Worldwide Quarterly Smartphone Tracker" of the International Data Corporation (IDC), as of the first quarter of 2022, the global smartphone penetration rate is approximately 82%) has ushered in the era of mobile Internet. In addition, with the development and wide use of the global positioning system (GPS), the generation and development of location-based services have become inevitable trends. Location-based services not only offer convenience but also provide new data sources and possibilities for business intelligence analysis, public affairs management, academic research, and other efforts. A large number of users around the world generate numerous information for sharing every day, and the shared information can be accessed via application interfaces [21]. The geographic data generated through smartphone location sharing services has brought a new revolution to GIS. The most fundamental issue involved in research using the shared data in mobile networks as a data source is the acquisition of data and the supporting platforms for the analysis and computation of acquired data. Population data estimation applies geo-fencing technology to push notifications. When a smartphone enters a geographical spatial scope, the location information of the smartphone can be acquired and logged. Since the size of a population is highly correlated to the number of smartphones, the current population size can be inferred by means of model simulation at different times and in different areas [22]. The volume of shared information in mobile networks is huge, which poses a great challenge in storing and analyzing this massive amount of data. Therefore, most of the research on mobile terminal location information focus on studying and predicting individual mobility patterns. Such analysis of individual mobility patterns, combined with the information from users' social network applications, can be suitably applied to public affairs management, such as user profiling, service recommendations, and market predictions [23]. Communication data providers can also upload smartphone location

data to cloud servers for real-time analysis and storage. Cloud servers can process large-scale location data, and the results of these cluster location data analyses can facilitate the development of more targeted emergency strategies in response to natural disasters. With the support of a large amount of data, the study of human activities on a larger scale will be of greater significance for urban and rural planning, population distribution, socio-economic indicators, and other aspects [24,25].

## Data preprocessing

A magnitude 6.8 earthquake struck Luding County, Sichuan Province (at 29.59 degrees north latitude and 102.08 degrees east longitude) at 12:52 p.m. Beijing time on September 5, 2022, with a focal depth of 16 kilometers. The earthquake caused heavy casualties, with 46 deaths, and severe damage to water, electricity, transportation, and communication facilities and other infrastructure. The region was highly deformed due to crustal movements, and there have been other violent earthquakes. Since 1900, 21 earthquakes with a magnitude of 6.0 or greater have occurred within 200 km of the epicenter, and a 6.2-magnitude earthquake occurred 27 km from the epicenter in 1975. At the epicenter, we collected a total of 1,806,100 smartphone location records and 4,856 spatial grid locations within a range of 300 km from east to west and 220 km from north to south, and the collection scope covers areas of VI degree and above.

The analysis in this study used GIS data is obtained from the open-source data Open Street Map (OSM), which is available for free and can be downloaded from the portal website of Digital Crete (https://www.openstreetmap.org/) The OSM data contains a series of data layers such as highways, railways, water systems, buildings, transportation facilities, etc. In this study, we only used data from residential points and areas, The DEM data adopts Copernicus DEM, which is a global open-source DEM data released by the European Space Agency(ESA) and can be downloaded from the ESA portal website (https://panda.copernicus.eu/panda). The DEM data of ESA has a 10 meter (EA-10) resolution for the European part and a 30 meter resolution for the global range. In this study, we used a 30 meter resolution in a raster format (Tif).The above data does not require authorization. The smartphone location records data obtained was authorized by the telecommunications company to the Zhejiang Earthquake Agency and provided to the author for use. The smartphone location records data used in this article uses Geohash encoding with an accuracy of Geohash7 (approximately 120m * 150m). This data is used to count the number of mobile devices in each Geohash grid within the earthquake zone range per minute.The data is group smartphone location data and does not involve the personal privacy of individual mobile phone users, so there is no concern about personal privacy leakage. All maps in this article are created using ARCGIS 10.6, The coordinates of the map are WGS84,The maps are oriented with North as up, and at this scale all maps in this article have an extent of 300km × 200km.The GIS data is obtained from open-source data websites and has been verified against the place name data of the epicenter area. All data does not involve copyright or legal disputes.

The epicenter of this earthquake was a mountainous area. The population density near the epicenter was not high, with only a few settlements and scenic areas for tourists. After the earthquake, the communication facilities in the disaster area were damaged, which led to a substantial reduction in mobile terminal connections. The disaster avoidance behavior of people and the rapid repair of mobile communication facilities also caused changes in the number of mobile terminals in the disaster area. In addition, there was also a huge quantitative difference between different time periods and areas. Therefore, we extracted the population information in different time periods. Specifically, starting from 10:00 a.m. on the 9th day, the data

coverage was extracted every 1 hour, with 24 time periods in total. In this way, the dynamic damage condition of communication facilities in the earthquake area could be reflected in a relatively comprehensive way. In this work, due to the vast data size, accuracy and efficiency contradicted each other. Although the adoption of high precision could contribute to a more detailed representation of population distribution, the problem of high computational complexity would occur. Moreover, the terrain of the disaster area was complicated, and the shadowing effect of the mountains had a certain impact on the accurate positioning of smartphone locations [26]. Therefore, an excessive pursuit of accuracy would result in a certain amount of repeated calculation points and affect the calculation efficiency. In order to analyze the population in the earthquake area in a faster way, while balancing accuracy and efficiency, we selected a 150 m grid size to analyze terminal location distribution.

### Population distribution simulation based on density analysis

**Principle of kernel density analysis.** Within the 150 m grid, the population is not perfectly uniformly distributed. Thus, a mathematical approach is needed to simulate population density. We regard the center of the grid as a point. The value of the point is the size of population in the grid, and the distribution of population density is represented by calculating point density [27]. There are three commonly used methods for calculating point density: the quadrat density method, the kernel density method, and the Voronoi diagram density method. The quadrat density method randomly selects a number of quadrats in the space of the area being simulated and calculates the density of each quadrat by counting the number of individuals in each quadrat, with the average of the density of all quadrats as the density of the large area. However, random sampling is characterized by a certain degree of subjectivity, so the simulation results are relatively larger. This method is applicable to the sampling survey of a static population, but it has a poor effect in simulating a population with strong mobility and high density. The Voronoi diagram density method calculates a distance-based plane partition in geometric space by using data points as generators of a Voronoi diagram. There are $n$ non-coincident seed points in the plane, and the plane is divided into $n$ regions in such a way that the distance from a point in each region to the seed point in the region in which the point is located is closer than the distance from it to any seed point in any other region, and each region is called a Voronoi seed point region. Due to the abrupt density changes at cell junctions and the neglect of continuity in the occurrence of spatial phenomena, Voronoi diagrams also have certain limitations in population distribution estimation [28,29].

However, the above problems can be solved by the kernel density method. The value of kernel density gradually decreases with increasing center radiation distance, with consideration to the distance attenuation effect of the center point on its surrounding locations [30]. Conceptually, each point is covered with a smooth curved surface, and the surface value is highest at the location of that point. As the distance increases, the surface value decreases until the value turns to zero at a distance equal to the search radius. Each pixel value of the output raster is the sum of all surface values superimposed on the pixel. Thus, the kernel density estimation method can transform a point set into a surface that exhibits continuous density variation. It also is possible to transform a discrete set of points into a smooth density variation diagram, thus demonstrating their spatial distribution pattern. The higher the density value, the greater the aggregation extent of the point is. Kernel density analysis has obvious advantages in the simulation of population distribution, as population distribution is featured with clustering, and the farther away from the center, the less dense the population distribution is [31].

**Spatial calculation method of kernel density.** In the calculation, the surface value is highest at the location of each point; the volume of the space formed by the curved surface and the

plane below is equal to the value of this point; and the density of each output raster pixel is the sum of the values of all kernel surfaces superimposed on the center of the raster pixel. The equation for the kernel density method is as follows:

$$f(s) = \sum_{i=1}^{n} \frac{1}{h^2} k\left(\frac{s - c_i}{h}\right) \tag{1}$$

where $f(s)$ is the kernel density calculation function for spatial location $s$; $h$ is the distance attenuation threshold; $n$ is the number of element points whose distance from location $s$ is less than or equal to $h$; and $k$ function refers to spatial weight function. The geometric significance of Eq (1) is that the density value is maximized at each core element $c_i$ and decreases as moving away from $c_i$ until the kernel density value decreases to 0 when the distance to the core element $c_i$ reaches the threshold $h$. There are 2 critical parameters in the kernel density function, i.e., the spatial weight function $k$ and the distance attenuation threshold $h$. Numerous studies have shown that the selection of spatial weight function has little effect on the results of point pattern distribution, while it is the choice of distance attenuation threshold that needs attention [32]. In practice, the setting of the threshold $h$ is mainly related to analysis scale and the characteristics of geographical phenomena. A smaller distance attenuation value can result in more high-value or low-value zones in the density distribution results, which is suitable for revealing the local characteristics of density distribution; a larger distance attenuation value enables hot spot regions to be more obvious under the global scale (Fig 2).

**Population distribution simulation based on kernel density analysis.** We adopted the kernel density method and used the collected smartphone point data to simulate the population distribution within a range of 300 km from east to west and 220 km from north to south. Figs 3 and 4 show the simulated population distribution of Luding based on smartphone data. As shown in Fig 3, the area near the epicenter is sparsely populated, with only a certain amount of population distributed in Moxi Town, Detou Town, and Dewei Township. In particular, the population density is very low in the west of the epicenter, with few large settlements located

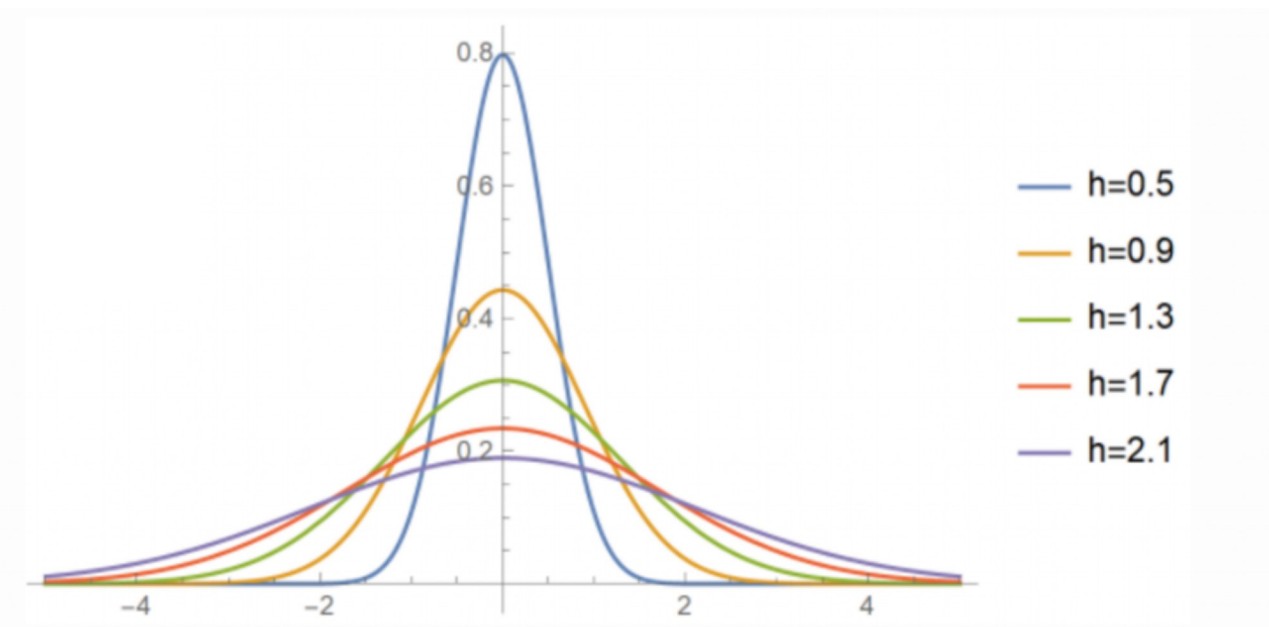

**Fig 2. Impact of the choice of the h parameter on the result.**

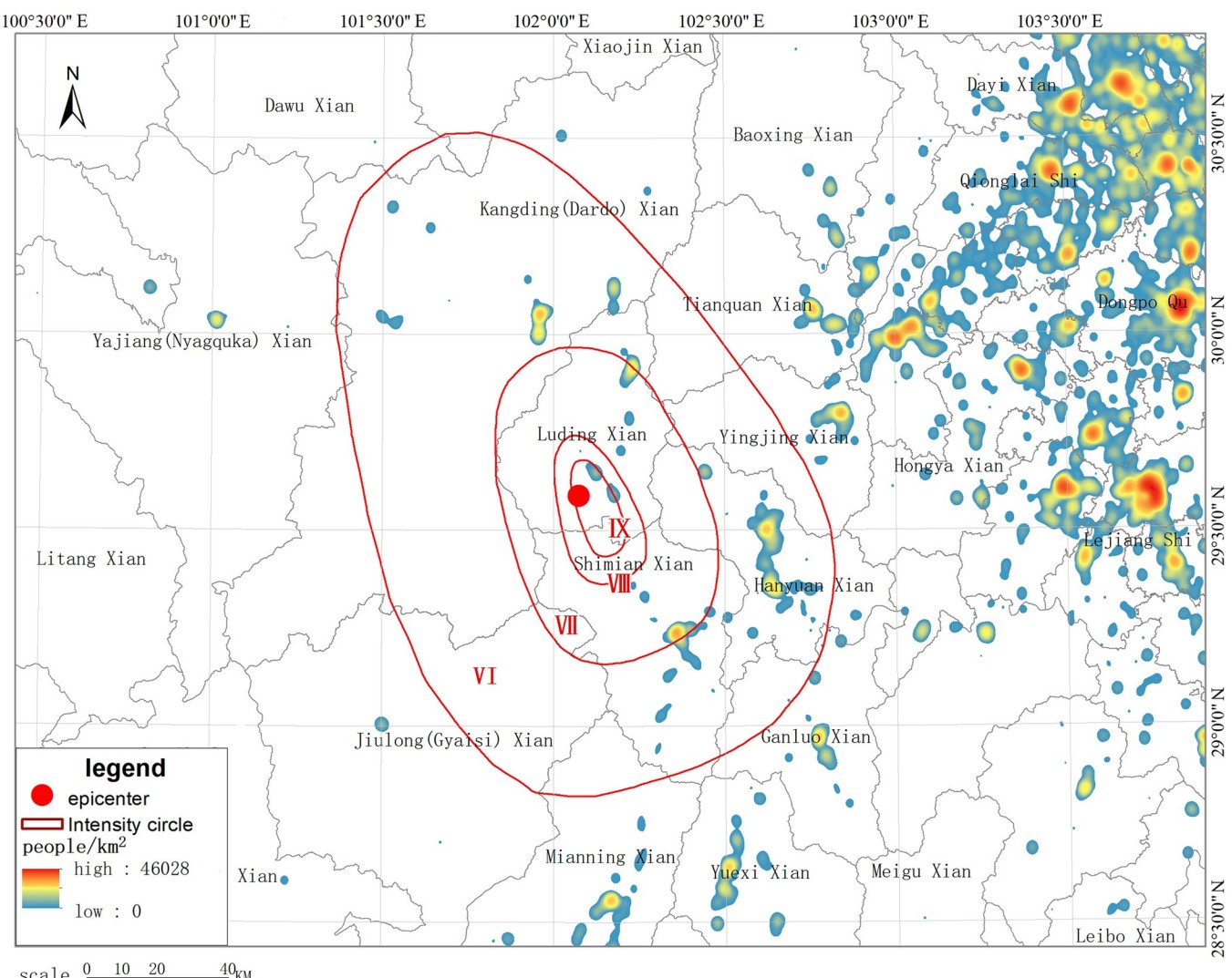

**Fig 3. Simulated pre-earthquake population distribution based on smartphone data.**

in the area. The population is concentrated in the eastern plain areas, which are far from the epicenter. Among these areas, Luding County is 40 km away from the epicenter, while Shimian County is 47 km away from the epicenter. The rest of the population is sporadically distributed along the bottom of the terrain ditches and the traffic lines. Fig 5 shows a heat map of population near the epicenter at 12:00 p.m. on September 5. It can be seen in Fig 5 that at 12:00 p.m., the area near the epicenter, where the government of Moxi Town is located, was densely populated, with only a few people sporadically distributed in the surrounding area. There were also few people in Detuo Town, and its neighboring township Dewei had a certain amount of population.

To reflect the population change before and after the earthquake, we compared the population density at 13:00 pm after the earthquake on September 5 (Fig 6) with the data at 12:00 pm before the earthquake (Fig 5) and calculated the density difference between the 2 time periods so that the communication outage and the movement of people caused by the earthquake can be reflected more objectively. Figs 5 and 6 show the changes in smartphone density before and after the earthquake in the earthquake area and the epicentral area, respectively. As shown in

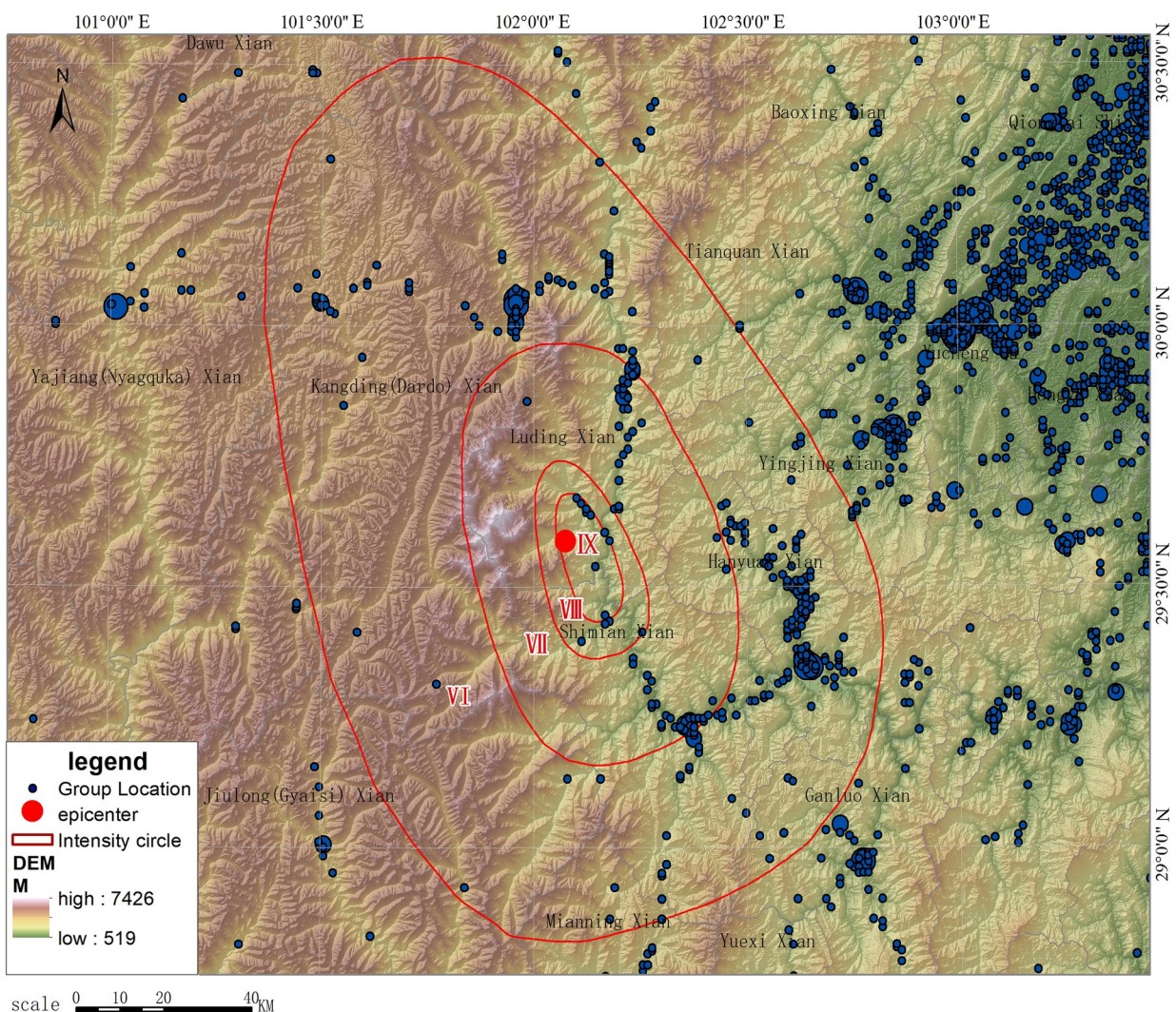

**Fig 4. Superimposing distribution of smartphone cluster location data and DEM.**

Fig 6, after the earthquake, the population density of highly seismic regions near the epicenter in the IX degree zone in the southern part of Luding County, such as Moxi Town, decreased significantly; as shown in Fig 7, the population density of areas farther away from the epicenter (such as Hanyuan County) did not change much, and the overall density decrease and increase of Xingjing County in the VI degree zone were relatively in balance. Yucheng District in Ya'an City, the most densely populated area, in the V degree zone, experienced an increase in the number of smartphones turned on. In the earthquake area of Luding County, the population density in Moxi Town, Detou Town, Yanzigou Town, Dewei Town and its surrounding area, as well as Wanggangping Township, Caoke Township and its surrounding area in Shimian County dropped sharply, from which it could be inferred that a significant number of out-of-service base stations and power outages occurred in these areas. Along National Highway 318 from Luding County to Tianquan County, there was a certain increase in population density, and the population on the periphery of Luding County also increased to a certain extent, indicating that tourists had already begun to evacuate out of the scenic areas half an hour after the earthquake.

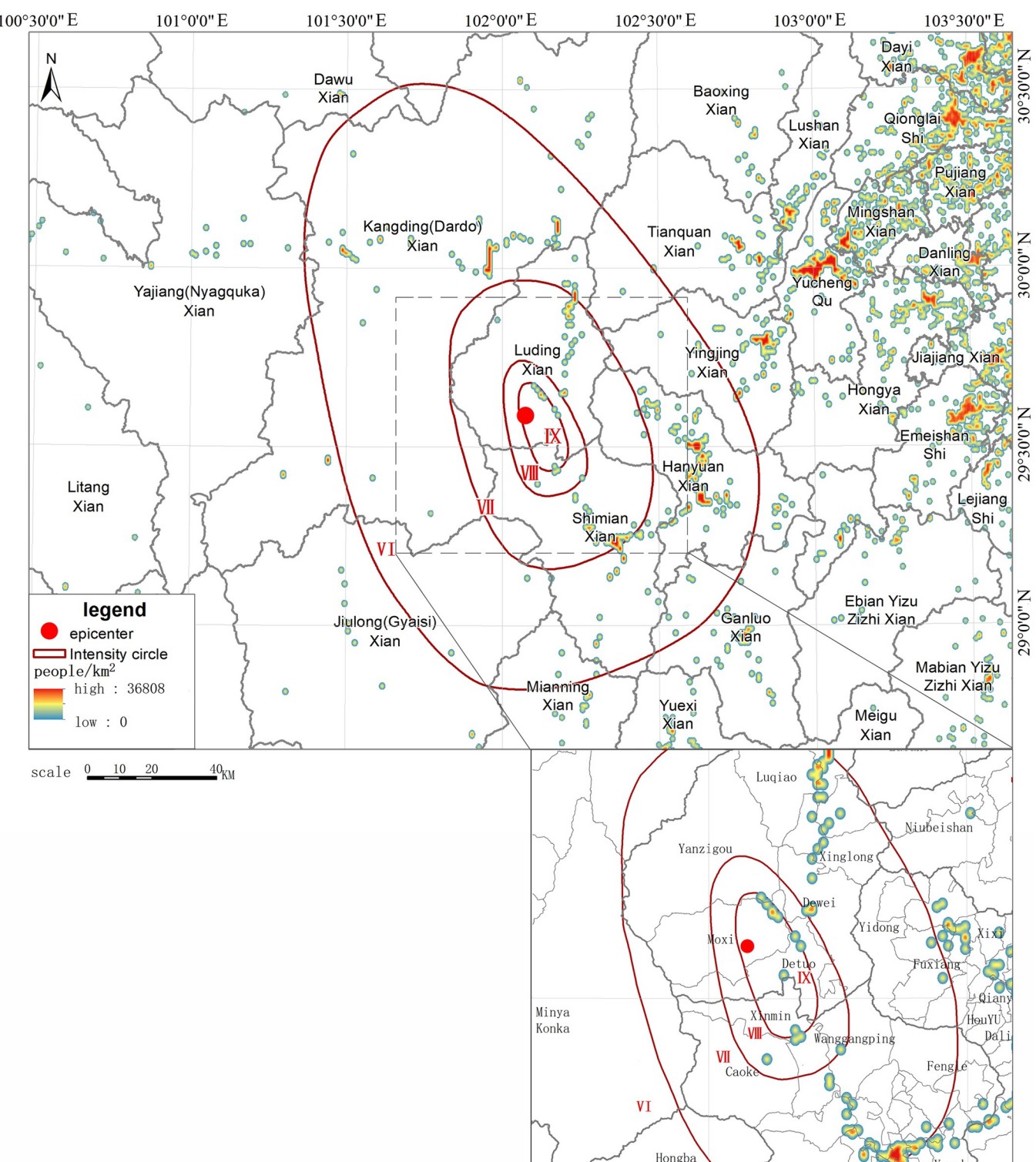

**Fig 5. Heat map of smartphone terminal distribution in the earthquake-stricken area before the earthquake.**

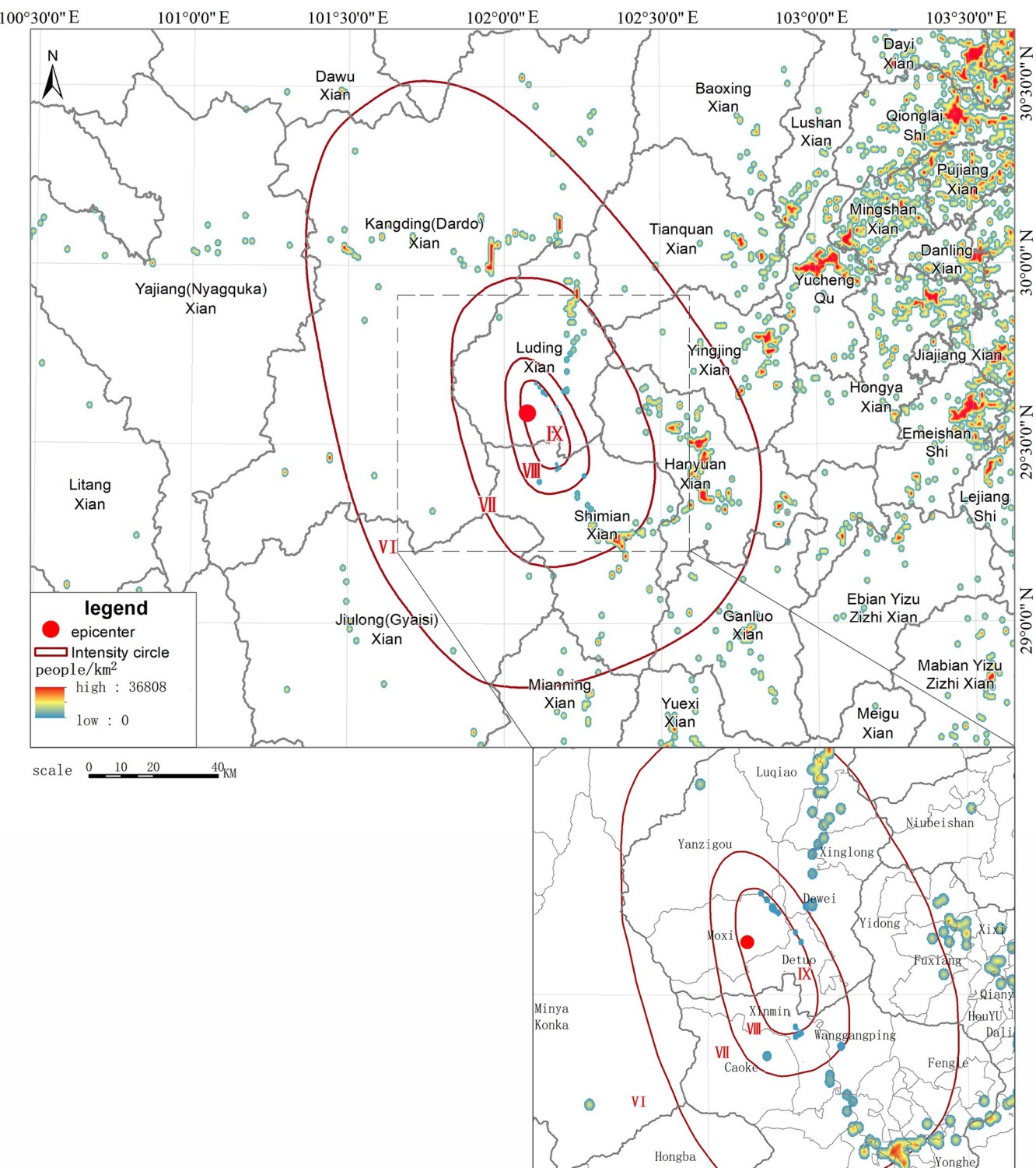

**Fig 6. Schematic diagram of changes in smartphone density after the earthquake.**

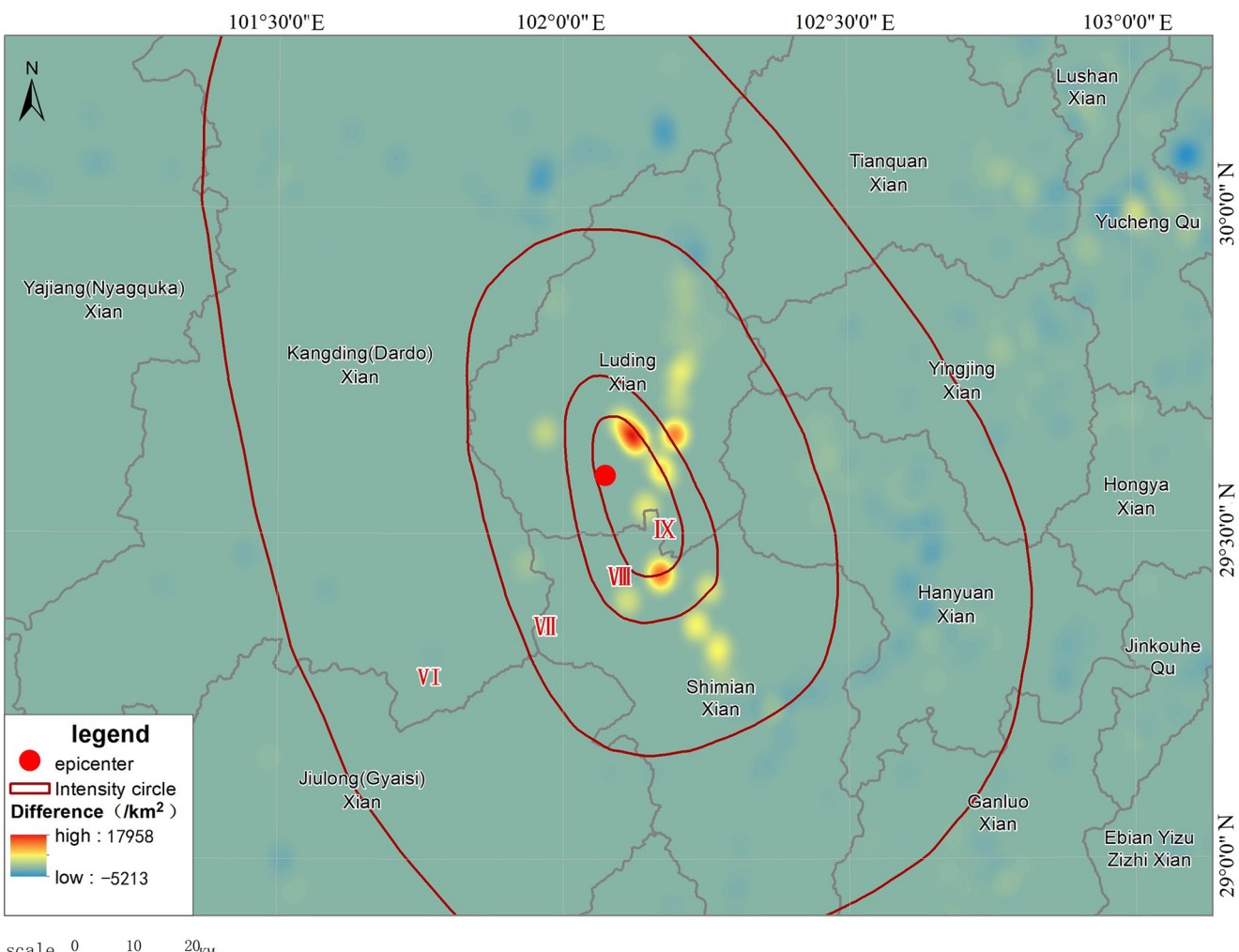

**Fig 7. Schematic diagram of changes in smartphone density in the epicentral area before and after the earthquake.**

## Results and discussion

### Analysis of population change in the segmented zones of the disaster area

After the earthquake, the population distribution in the disaster area displayed a dynamic change. Starting from the size of population affected by out-of-service base stations after the earthquake, to the repair of communication and electric lines, and then to the subsequent evacuation to the outside along the traffic lines, the number and location of the population had been changing.

We selected several typical settlements in the IX and VII degree zones for the time series analysis of communication terminal volume. Moxi Town, Caoke Township, and Detou Town in the IX degree zone were chosen, of which Moxi Town was closer to the epicenter and was also one of the settlements heavily struck by the earthquake. Fig 8 shows an analysis of the population data by area and time period within 24 hours since 12:00 pm on September 5. At 1:00 pm, communication and power systems were severely damaged after the earthquake, resulting in a sharp drop in communication terminal volume. However, the communication repair was made very quickly. After the emergency communication vehicles entered the earthquake area,

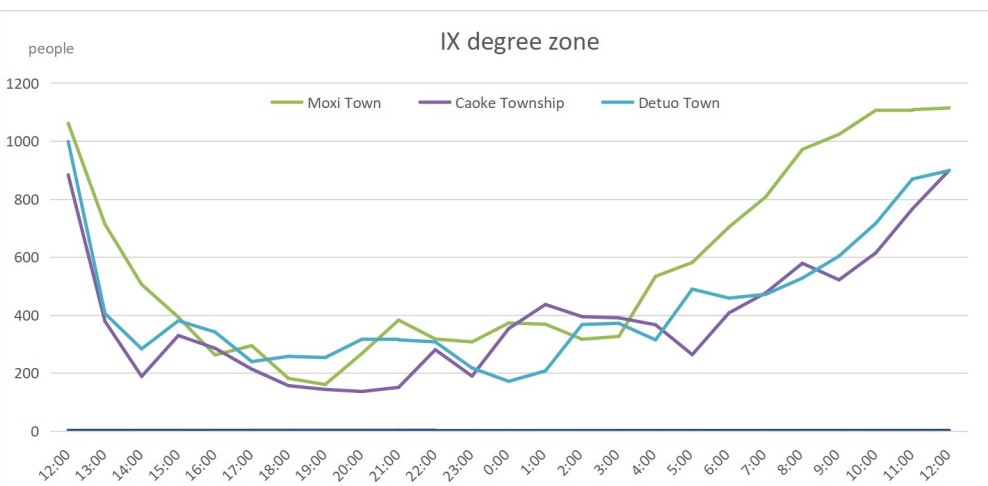

**Fig 8. Time series change of population in settlements in IX degree zone.**

communication was partially restored. As of 7:00 pm that night, part of the communication and power systems had been restored. The communication terminal volume began to grow slowly. At 06:00 am on September 6, the population evacuated from the scenic areas and the rescue forces entering the disaster area were superimposed. As shown in Fig 8, the crowd flow in the three settlements showed a sharp increase; the situation in Caoke Township and Detou Township was basically similar, and the time series curves of the communication terminal volume in the two places were highly consistent.

We also examined statistics on changes in Shimian County, Luding County, and Yidong Town in the VII degree zone (Fig 9) within 24 hours. Because of the extremely uneven population distribution in this zone, in which Shimian Town and Luding County Town were relatively densely populated and the damage to communication facilities after the earthquake was less serious than that in the IX degree zone, the decline in communication terminal volume was relatively slower than that in the other zones. At 11:00 pm, the communication terminal volume showed a natural decline due to equipment shutdowns. Similarly, the population change in the three typical settlements in the VII degree zone is highly consistent. What is

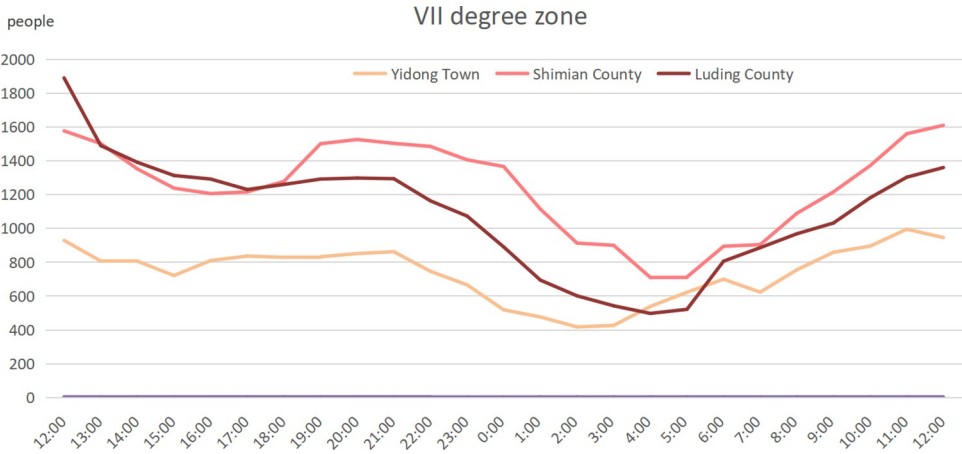

**Fig 9. Time series change of population in settlements in VII degree zone.**

different from the IX degree zone is that there was a continuous outflow of people after the partial restoration of communication.

## Simulation of out-of-service base station situations in the disaster area

In previous severe earthquakes, there has always been a phenomenon of out-of-service mobile communication base stations in the hardest hit areas. For example, the 8.0-magnitude earthquake in Wenchuan in 2008 caused 3,429 mobile communication base stations of China Mobile Group Sichuan Sub-Company to be out of service; the 7.1-magnitude earthquake in Yushu in 2010 caused the out-of-service rate of local mobile communication base stations to reach over 75%; the 7.0-magnitude earthquake in Lushan in 2013 caused 303 base stations to be out of service; and in the 7.0-magnitude earthquake in Jiuzhaigou in 2017, 235 mobile communication base stations were out of service [33]. After this Luding Ms 6.8 earthquake, the highest intensity in the hardest hit area was IX degree, and there were also communication interruptions. In the disaster area, a large number of base stations were out of service, and fiber optic cables were disrupted (Sichuan Communications Administration, 2017). By comparing the reduction in smartphone-based population before and after the earthquake, we could estimate the out-of-service rate and distribution range of base stations. We divided the earthquake area into 250×250 m grids and counted the variation $R$ in population in each grid before and after the earthquake:

$$R = (N-K)/N \tag{2}$$

where $N$ refers to the size of population before the earthquake; $K$ refers to the size of population after the earthquake; and $R$ is the out-of-service rate. In the actual calculation (Fig 10), there would be a post-earthquake increase in the number of smartphones in individual grids due to the aggregation of people, statistical errors, etc., but the overall number of smartphones gradually decreased from the epicenter to the outside. The calculated maximum value of $R$ is 81.59%, which mainly occurred around Moxi Town. This indicates that the area was hit hardest. Part of the area might experience service outages for all base stations. However, as smartphones can still provide location information through other communication means such as satellite, WIFI, Bluetooth, etc., the out-of-service rate would generally not reach 100%. During the mapping process, we excluded grids with $R<0$, i.e., those with increased population, to facilitate analysis and distributed the increased values evenly to the surrounding neighboring grids to overlay with the final intensity survey results. Fig 11 shows the simulated distribution of out-of-service base stations based on the change in the number of smartphones after eliminating interference. The simulated results are very close to those obtained from the later field investigation, that is, grids with more than a 50% reduction in smartphones are largely located in the VII degree zone, with an overall trend of slightly extending to the northwest. The densely populated areas in the east that are farther from the epicenter show a general slight increase in the number of smartphones, which coincides with the increase in communications after the earthquake as the surrounding population started to express concern about the disaster. Because there are mountainous areas to the west of the epicenter, with an elevation of more than 3,000 m and a sparse population (see Fig 4), a large empty space was shown in the mobile terminal locations we obtained. After interpolation, the simulated interpolation data of the intensity distribution were greatly distorted compared with the real situation due to the density of the data points being too small (Fig 10). In general, the intensity distribution is roughly spread outward with the epicenter as the center. Here, we shifted the values of the two symmetrical points on the east side of the epicenter to the west, took them as interpolation control points (Fig 11) to fill in the blank values, and finally achieved a relatively ideal effect.

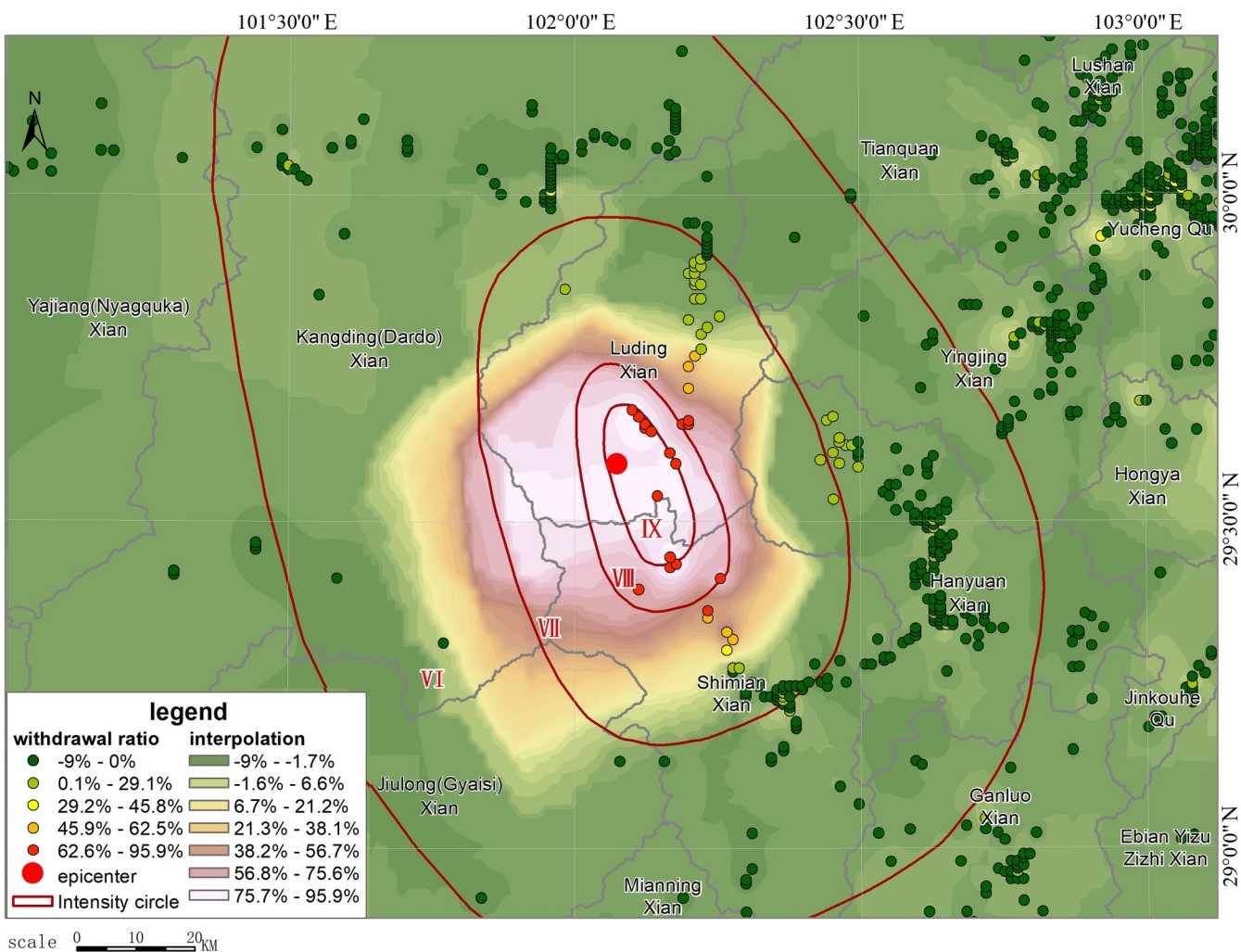

**Fig 10. Disaster area distribution interpolation diagram based on the change in the number of smartphones (raw data).**

These data have generated a good simulation of out-of-service base stations, so it is entirely possible to infer the out-of-service base station situation from the reduced percentage of the smartphone-based population, thereby inferring the distribution of the hardest hit areat (see S1 File).

After the intensity map was officially released, we also calculated the changes in simulated population based on smartphones in different intensity zones before and after the earthquake, and in order to exclude the impact of smartphone shutdowns for lunch breaks on the out-of-service situation, we also calculated the changes in the smartphone-based population in the same time period and the same area on September 4. In the calculation of the final result, as the base, the data on September 4 was subtracted so that we could obtain the real out-of-service situation on September 5. Table 1 shows the statistics of the out-of-service rate based on smartphone population data. In Table 2, it can be seen that the out-of-service rate in the VI degree zone is -1.62%, while the out-of-service rate on September 4 is 1.32%. The out-of-service rate decreased after the earthquake, which means that after the earthquake, some people in this zone were affected and kept learning about the disaster through their mobile terminals. By using the same method and after deduction of the base, we finally calculated that the out-of-

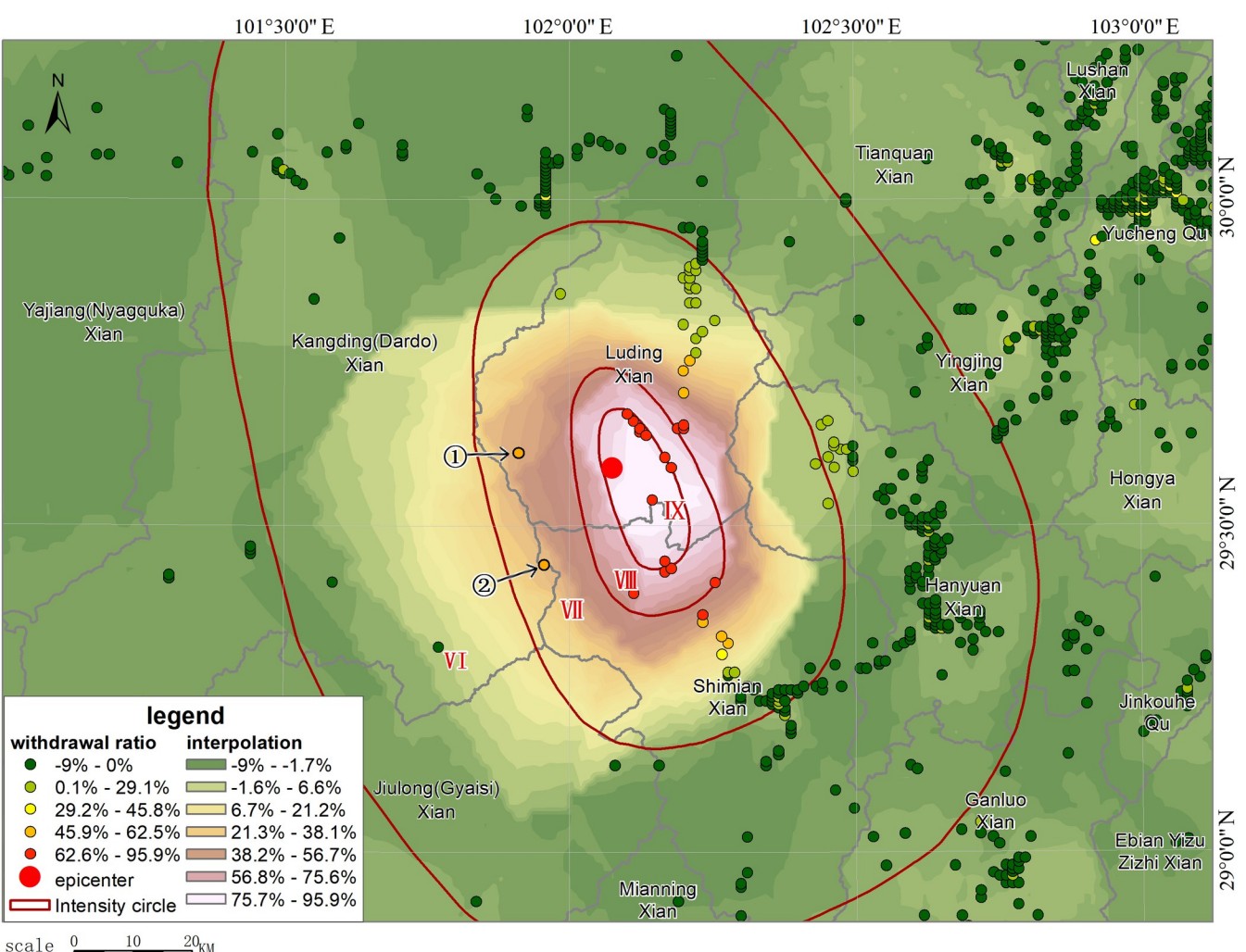

**Fig 11. Disaster area distribution interpolation diagram based on the change in the number of smartphones after correction by adding control points.**

service rate in the VII degree zone at 19.42%, which is contrary to the traditional view that service outages at communication base stations will occur only in the VIII degree zone; thus, it is inferred that there exists a certain out-of-service rate in the VII+ degree zone. In the VIII degree and IX degree zones, we believe that the number of people who were strongly affected by the earthquake and turned off their smartphones at 1:00 pm was small. Thus, the final results of the out-of-service rate in VIII degree and IX degree zones are 48.40% and 78.11%. The reduction in the number of smartphones after the earthquake may be affected by multiple

**Table 1. Correction of out-of-service rate based on the previous day's mobile terminal location data.**

| S/N | Intensity | Population at 1:00 pm on September 4 | Population at 1:00 pm on September 5 | Out-of-service rate at the same time on September 4 (%) | Out-of-service rate on September 5 (%) | Corrected out-of-service rate (%) |
|---|---|---|---|---|---|---|
| 1 | VI | 94563 | 95811 | 1.32 | -1.62 | -3.52 |
| 2 | VII | 21414 | 22166 | 3.51 | 22.26 | 19.42 |
| 4 | VIII | 17271 | 17055 | -1.25 | 51.06 | 48.40 |
| 3 | IX | 5066 | 5191 | 2.47 | 81.59 | 78.11 |

**Table 2. Statistics on out-of-service rate based on mobile terminal location data in previous major earthquakes.**

| Earthquake | Data | Magnitude | Out-of-service rate/% | | |
|---|---|---|---|---|---|
| | | | VII | VIII | IX |
| Menyuan, Qinghai | 8-Jan-22 | 6.9 | 20.03 | 54.35 | 82.09 |
| Lushan, Sichuan | 1-Jun-22 | 6.1 | 22 | 55.87 | |
| Luding, Sichuan | 5-Sep-22 | 6.8 | 19.4 | 48.4 | 78.1 |
| Biru, Tibet | 19-Mar-21 | 6.1 | 17.24 | 39.34 | |
| Maduo, Qinghai | 22-May-21 | 7.4 | 22 | 46.8 | 87.35 |
| Payzawaty, Xinjiang | 19-Jan-20 | 6.4 | 21 | 39.6 | |
| Yutian, Xinjiang | 26-Jun-20 | 6.4 | 19.5 | 38.7 | |
| Jiuzhaigou, Sichuan | 8-Aug-17 | 7 | 27 | 45.4 | 69.24 |
| Xayary, Xinjiang | 30-Jan-23 | 6.1 | 19 | 40 | |
| Changning, Sichuan | 17-Jun-19 | 6 | 24 | 53.9 | |
| Milin, Tibet | 18-Nov-17 | 6.9 | 19 | 50 | |

factors, but an out-of-service base station is the most important one. Therefore, the above results can objectively reflect the base station out-of-service rate in different intensity zones.

With the same method, we tracked back to the population heat map data of earthquakes with a magnitude of 6.0 or above in mainland China since 2017 in the database, analyzed and processed the communication location big data in the zones of VII degree or above during multiple post-earthquake time periods in the course of earthquake emergency response, used the spatial change of terminal data to infer the damage of communication base stations and the response of people, and summarized the regression modeling so that we can spatially infer the distribution range of the hardest hit areas. Based on the change trend of terminal location data, the research proceeded with the extraction of seismic damage information to correct the empirical isoseismal line so as to make up for the lack of first-hand field data of the existing rapid assessment system. Finally, a rapid assessment method of the seismic influence field based on communication big data is established, and an operable software system is formed, which can provide technical support for earthquake emergency rescues. At present, the related research results have been popularized and applied in many regions.

The exponential function of Eq (3) is used to conduct fitting on the relation between the magnitude of previous earthquakes and the post-earthquake out-of-service rate of base stations. The fitted curves and the actual data points are shown in Fig 12, and a goodness of fit of the fitted curves is obtained: R = 0.68.

$$P = 3.7M^2 - 12.3M + 3.2 \qquad (3)$$

where P is the base station out-of-service rate and M is the intensity. The polynomial fitting model can better accommodate the nonlinear relationship of the data. In Fig 12, the distribution of data points is concentrated and shows a clear linear trend. The use of polynomial fitting can better capture the characteristics of the data and provide a relatively flat trend line to describe the relation between out-of-service rate and intensity. This model features with a high degree of flexibility and can better simulate data changes. Therefore, the polynomial fitting model is an appropriate choice in this context.

## Conclusions

With good timeliness and high data accuracy, smartphone population heat map data can be applied to the actual practice of earthquake emergency response. Taking the Luding Ms 6.8 earthquake as an example, we have obtained the simulation data of the population distribution

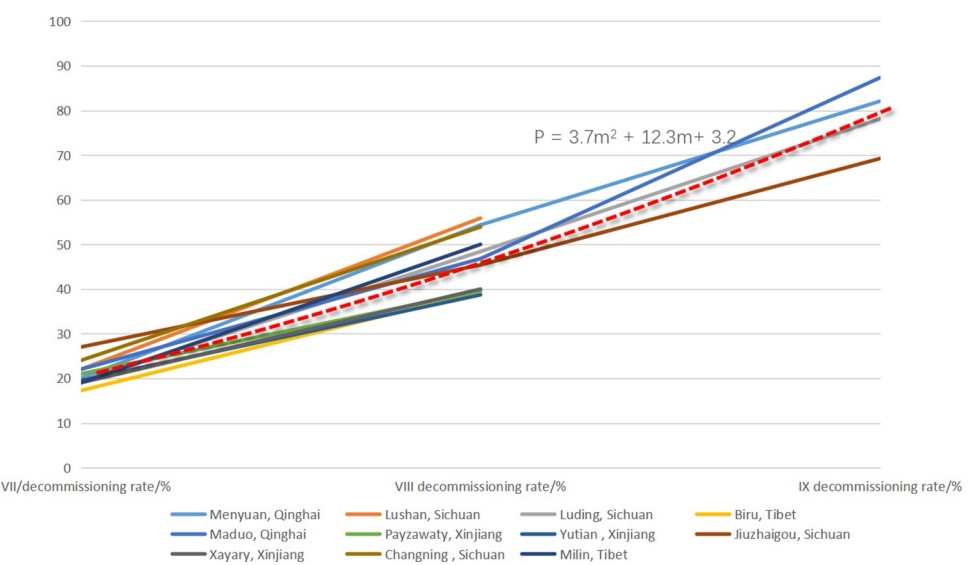

**Fig 12. Base station out-of-service rate and intensity relation fitting curves in previous earthquakes.**

in the earthquake area in real time with the support of smartphone location data. Having been tested by dozens of large and small earthquakes, the data model has become more mature. Based on the cross-corroboration of multiple validation channels, we find that the data obtained at this time is consistent with the actual situation. In the hours after an earthquake, smartphone location data can provide strong support for the government's disaster relief work when the actual disaster situation is still uncertain. The application of this method in the emergency management field is also widely recognized. Compared with traditional modes of disaster information acquisition, smartphone location data has obvious advantages in terms of cost, accuracy, efficiency, and other aspects. With high timeliness and good continuity of data, as well as the absence of additional investment in hardware equipment and organization of large-scale field investigations, this method can realize the rapid acquisition of the location change rule of communication terminal population in a disaster area after an earthquake. By analyzing the characteristics of spatial–temporal changes in smartphone location big data before and after the earthquake, we can infer the strength and distribution range of earthquake intensity. In addition, unlike the "black box period" in previous earthquakes, this work enables us to provide the government with a sufficient and reliable information basis for disaster relief within one hour after the earthquake, even when a more accurate and complete picture of the disaster situation is not available. This approach can greatly complement the shortcomings of the existing rapid seismic assessment systems and enables the rapid obtainment of highly credible disaster information in a timely manner when a sudden and destructive earthquake occurs (Table 3).

An out-of-service base station will cause a large amount of mobile terminal location data to disappear after an earthquake. During this earthquake, the magnitude reached 6.8, the maximum intensity of the epicenter was IX degree, and the hardest hit area experienced power interruptions and out-of-service base stations, which directly led to a cliff-like drop in the mobile terminal location data we obtained after the earthquake. The closer to the epicenter, the more obvious this phenomenon was. After the communication and power lines were repaired, the acquisition of smartphone location data began to recover gradually. Determining the base station out-of-service rate plays an important indicative role in estimating the seismic intensity of the hardest hit area in the first moments. This further helps us to obtain the seismic

**Table 3. Comparison of main earthquake emergency assessment methods and indicators.**

| Clauses | Rapid earthquake disaster assessment method based on multi-source communication data | Earthquake emergency intensity analysis based on remote sensing data | Influence field calculation method based on empirical model |
|---|---|---|---|
| Main data status | Based on communication big data, it is a direct feedback of the original data | Satellite remote sensing data | Geographic information data, seismic structure data, economic population data, etc., have a certain lag |
| Response time | 10 minutes after the earthquake (theoretical value) | 2–24 hours | 30 minutes |
| Assessment accuracy | Minimum 20 meter | 0.5 meters—30 meters | The township administrative region |
| Assessment data timeliness | 10-minute update | Limited by transit satellite time | Manual renewal (Annual statistics) |
| Assessment error | Approaching the actual investigation results | There will be significant errors | There will be significant errors |
| Reliability of intensity assessment | high | Middle | Middle |
| Defect | There is a certain deviation in the thermal data of mobile phones in different regions after the earthquake | The relationship and mechanism between the characteristics of image earthquake damage and the mode of disaster formation are not yet perfect | Constrained by active faults and site condition data |

intensity, isoseismal line, and other critical information in the affected areas. With the support of communication big data, we can correct the empirical isoseismal lines based on the seismic damage information extracted from the model. In addition, we can also verify the locations that cannot be accurately assessed and dynamically modify the isoseismal lines to gradually improve the assessment accuracy. By combining empirical models and automatic computer processing technologies, we can obtain the earthquake disaster information in a faster manner. In the meantime, we still need to discover the characteristics of seismic damage in different regions and of different magnitudes, and on this basis, we can establish the basis of determining seismic intensity according to the performance of smartphone location data in different situations. This will ultimately further improve the method of obtaining the seismic intensity influence field based on mobile Internet data and enhance our earthquake response capability.

In practice, we have also found some shortcomings in using mobile terminals to judge disaster situations. Even in modern society, the utilization rate of smartphones is still closely related to factors such as age, geographic location and economic status. Large-intensity earthquakes usually occur in mountainous and sparsely populated areas. In the disaster area of this Luding earthquake, there was also an extreme situation where the population in the mountainous area in the northwest of the disaster area was extremely small and there was a clear gap in population composition, as most of the population were aged over 60 or children, with low smartphone usage rates, which may have led to underestimated location data. In contrast, Moxi Town and Detou Town near the epicenter were relatively densely populated. As Moxi Town is a scenic location, most of the population here were tourists, with a high smartphone usage rate, which may have led to overestimating the pre-earthquake data to a certain extent if a unified model was used. Therefore, in order to estimate and judge disaster situations more accurately, it is necessary to customize a suitable regional population model to take these differential factors into account. In the Luding Ms 6.8 earthquake, the research team applied different population models to adapt to the special conditions in Moxi Town and the northwestern mountainous areas. This has fully demonstrated the importance of flexibility and customization strategies. This approach can not only make up for the data bias caused by uneven smartphone usage rate but also help improve the accuracy of disaster assessment.

Smartphone-based location big data has great application potential in future earthquake emergency management. This technology can not only realize the real-time estimation of population distribution during an earthquake but also achieve in-depth analysis based on its rich attribute information, such as tracking the places of origin of the population, the population movement vector, traffic jams, etc., which could not be accomplished in past earthquake emergency responses. The mining of smartphone data can derive richer applications to better serve earthquake emergency responses. The application of mobile location big data sources can replace the traditional field investigation of seismic damages. We can calculate the spatial distribution of a disaster based on the disaster information reflected in big data so as to obtain the simulated results of disaster distribution, which will greatly improve the efficiency of seismic damage information acquisition.

## Supporting information

**S1 File. Calculation on out-of-service rate based on mobile terminal location data in this earthquakes.**
(ZIP)

## Author Contributions

**Conceptualization:** Huanyu Li.

**Data curation:** Qingquan Tan, Jingfei Yin, Haiqing Sun.

**Formal analysis:** Dongping Li, Qingquan Tan, Jingfei Yin.

**Methodology:** Qingquan Tan.

**Resources:** Zhiyi Tong, Min Li.

**Software:** Dongping Li, Zhiyi Tong, Min Li.

**Supervision:** Zhiyi Tong, Min Li.

**Writing – original draft:** Dongping Li.

**Writing – review & editing:** Dongping Li.

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
