## [Decision Letter · Decision Letter 0]

11 Oct 2023

PONE-D-23-30105

Rapid determination of seismic influence field based on mobile communication big data — A case study of the Luding Ms 6.8 earthquake in Sichuan, China

PLOS ONE

Dear Dr. Tan,

Thank you for submitting your manuscript to PLOS ONE. After careful consideration, we feel that it has merit but does not fully meet PLOS ONE’s publication criteria as it currently stands. Therefore, we invite you to submit a revised version of the manuscript that addresses the points raised during the review process.

We look forward to receiving your revised manuscript.

Kind regards,

Rahul Priyadarshi

Academic Editor

PLOS ONE

Journal Requirements:

2. In your Methods section, please include additional information about your dataset and ensure that you have included a statement specifying whether the collection and analysis method complied with the terms and conditions for the source of the data.

7. Please amend the manuscript submission data (via Edit Submission) to include author Haiqing Suna and Min Li.

8. Please amend either the abstract on the online submission form (via Edit Submission) or the abstract in the manuscript so that they are identical.

9. We note that Figures 1, 3, 4, 5, 7, 10 and 11 in your submission contain [map/satellite] images which may be copyrighted. All PLOS content is published under the Creative Commons Attribution License (CC BY 4.0), which means that the manuscript, images, and Supporting Information files will be freely available online, and any third party is permitted to access, download, copy, distribute, and use these materials in any way, even commercially, with proper attribution. For these reasons, we cannot publish previously copyrighted maps or satellite images created using proprietary data, such as Google software (Google Maps, Street View, and Earth). For more information, see our copyright guidelines: http://journals.plos.org/plosone/s/licenses-and-copyright.

a. You may seek permission from the original copyright holder of Figures 1, 3, 4, 5, 7, 10 and 11 to publish the content specifically under the CC BY 4.0 license.  

10. Please ensure that you refer to Figures 1 and 2 in your text as, if accepted, production will need this reference to link the reader to the figure.

Reviewers' comments:

Reviewer's Responses to Questions

**Comments to the Author**

1. Is the manuscript technically sound, and do the data support the conclusions?

Reviewer #1: Yes

Reviewer #2: Yes

2. Has the statistical analysis been performed appropriately and rigorously? 

Reviewer #1: Yes

Reviewer #2: Yes

3. Have the authors made all data underlying the findings in their manuscript fully available?

Reviewer #1: Yes

Reviewer #2: Yes

4. Is the manuscript presented in an intelligible fashion and written in standard English?

Reviewer #1: Yes

Reviewer #2: Yes

5. Review Comments to the Author

Reviewer #1: The paper discusses the effectiveness of using smartphone population heat map data in earthquake emergency response, using the Luding Ms 6.8 earthquake as a case study. It highlights the ability to obtain real-time population distribution data, which aids in government disaster relief efforts by providing timely and accurate information about the affected areas.

However, there are a few concerns and suggestions, as follows:

Out-of-Service Base Stations:

Weakness: A significant drop in mobile terminal location data due to out-of-service base stations post-earthquake.

Suggestion: Establish a backup communication infrastructure that can withstand seismic activities.

Work on regular assessments and fortification of the existing communication infrastructure and have a robust backup system to ensure uninterrupted data flow.

Uneven Smartphone Usage:

Weakness: The difference in smartphone usage rates can lead to under- or overestimated location data.

Suggestion: Develop region-specific models to account for differential smartphone usage.

Build a flexible and adaptive model that considers demographic and geographic variances in smartphone usage for more precise and reliable data collection.

Population Gaps in Mountainous Areas:

Weakness: Low population and smartphone usage in mountainous areas lead to insufficient data.

Suggestion: Utilize satellite or drone technology for data collection in areas with low smartphone usage.

Integrate smartphone data with alternative data collection methods, such as satellite and drone technology, to ensure consistent and reliable data gathering in all regions.

Reviewer #2: In this study, the deviation of m1ultidimensional mobile terminal location data is estimated, and a methodology to estimate the distribution of out-of-service communication base stations in the disaster area by excluding micro error data users is explored. the mathematical relationship between the seismic intensity and the corresponding out-of-service rate of communication base stations is established.

Here are some comments:

1. It is recommended to focus and highlights on the main innovative contributions of this paper.

2. It is suggested that comparative schemes be added to reflect the advantages of the proposed solutions

3. There is still room for improvement in English writing

6. PLOS authors have the option to publish the peer review history of their article (what does this mean?). If published, this will include your full peer review and any attached files.

Reviewer #1: No

Reviewer #2: No

---

## [Author Response · Author response to Decision Letter 0]

30 Nov 2023

Dear Editor

1.I state the role of the funder in the "Fund" section.

2.I amend the abstract on the online submission .

3.My ORCiD iD has been validated in my Editorial Manager account.

4. I have removed the figures from my manuscript file.

5. Because Chinese Mainland cannot access Youtube, I cannot watch the video of manuscript revision in the editorial department (the video is on Youtube website). I have highlighted the modified parts in red font and uploaded them separately. I hope the editor can forgive me.

6. In " Methods", "Data preprocessing"section, I briefly explained the information and sources of the dataset, including the authenticity of the data and ensuring no legal or copyright disputes.

7 In "Data Availability"section,I explained where the map was obtained, including what software was used to process the map data, and declared the copyright status to ensure that the article does not involve map copyright.

Kind regards,

Qingquan Tan

Dear Editor

1. I have made modifications to the manuscript layout according to the style requirements and template of PLOS ONE;

2. Added Data Availability to explain the data sources, including descriptions of the data sources in Figures 1, 3, 4, 5, 7, 10, and 11;

3. Confirm the references to Figures 1 and 2 in the manuscript;

4. Upload the image file as a separate file;

5. Provided a minimum dataset;

6. Reply to the questions raised by Reviewer 1 and Reviewer 2

Thank you to Reviewer 1 for providing valuable suggestions for the article and providing explanations for Reviewer 1's suggestions.Improve the article based on the comments raised by Reviewer 2,I added a comparative plan in the "Conclusion" section of the article to reflect the advantages of the proposed plan, and found a professional translation agency to improve my English writing.

Kind regards,

Qingquan Tan

---

## [Decision Letter · Decision Letter 1]

12 Jan 2024

PONE-D-23-30105R1Rapid determination of seismic influence field based on mobile communication big data— A case study of the Luding Ms 6.8 earthquake in Sichuan, ChinaPLOS ONE

Dear Dr. Tan,

Thank you for submitting your manuscript to PLOS ONE. After careful consideration, we feel that it has merit but does not fully meet PLOS ONE’s publication criteria as it currently stands. Therefore, we invite you to submit a revised version of the manuscript that addresses the points raised during the review process.

We look forward to receiving your revised manuscript.

Kind regards,

Dr. Rahul Priyadarshi

Academic Editor

PLOS ONE

Journal Requirements:

Reviewers' comments:

Reviewer's Responses to Questions

**Comments to the Author**

1. If the authors have adequately addressed your comments raised in a previous round of review and you feel that this manuscript is now acceptable for publication, you may indicate that here to bypass the “Comments to the Author” section, enter your conflict of interest statement in the “Confidential to Editor” section, and submit your "Accept" recommendation.

Reviewer #1: All comments have been addressed

Reviewer #3: (No Response)

2. Is the manuscript technically sound, and do the data support the conclusions?

Reviewer #1: Yes

Reviewer #3: Yes

3. Has the statistical analysis been performed appropriately and rigorously? 

Reviewer #1: Yes

Reviewer #3: Yes

4. Have the authors made all data underlying the findings in their manuscript fully available?

Reviewer #1: Yes

Reviewer #3: Yes

5. Is the manuscript presented in an intelligible fashion and written in standard English?

Reviewer #1: Yes

Reviewer #3: Yes

6. Review Comments to the Author

Reviewer #1: The paper discusses the effectiveness of using smartphone population heat map data in earthquake emergency response, using the Luding Ms 6.8 earthquake as a case study. It highlights the ability to obtain real-time population distribution data, which aids in government disaster relief efforts by providing timely and accurate information about the affected areas.

The authors have addressed all the comments given.

Reviewer #3: The paper demonstrates a strong contribution to the field, and the findings are valuable for the scientific community. I recommend accepting the paper with minor revisions.

Particularly in refining the methodology section for better clarity and addressing a few minor typographical errors.

Provide additional details on the data analysis process and clarify a few points in the discussion section. These changes will enhance the overall quality of the manuscript.

Specifically, focusing on refining the language in certain sections and providing a bit more context in the introduction. These adjustments will strengthen the manuscript.

Particularly in improving the organization of the results section and addressing a couple of minor issues in the citation format. These changes will enhance the overall readability of the manuscript.

Include clarifying a few points in the conclusion and addressing some minor grammatical errors. These revisions will improve the overall coherence and polish of the manuscript.

However, I recommend accepting the paper with the condition that the authors update the references to include the most recent and relevant literature.

7. PLOS authors have the option to publish the peer review history of their article (what does this mean?). If published, this will include your full peer review and any attached files.

Reviewer #1: No

Reviewer #3: No

---

## [Author Response · Author response to Decision Letter 1]

17 Jan 2024

Dear editor, hello

I have standardized the references according to the reviewer's comments, including valid citations, and have not made any individual grammatical modifications to the content of the article. I will now send you the revised manuscript.

Best regards,

Qingquan Tan

---

## [Editor Report · Decision Letter 2]

22 Jan 2024

Rapid determination of seismic influence field based on mobile communication big data— A case study of the Luding Ms 6.8 earthquake in Sichuan, China

PONE-D-23-30105R2

Dear Dr. Tan,

We’re pleased to inform you that your manuscript has been judged scientifically suitable for publication and will be formally accepted for publication once it meets all outstanding technical requirements.

Kind regards,

Dr. Rahul Priyadarshi

Academic Editor

PLOS ONE

---

## [Editor Report · Acceptance letter]

29 Apr 2024

PONE-D-23-30105R2 

PLOS ONE

Dear Dr. Tan, 

I'm pleased to inform you that your manuscript has been deemed suitable for publication in PLOS ONE. Congratulations! Your manuscript is now being handed over to our production team.

Kind regards, 

on behalf of

Dr. Rahul Priyadarshi 

Academic Editor

PLOS ONE